# The Roles of *SNHG* Family in Osteoblast Differentiation

**DOI:** 10.3390/genes13122268

**Published:** 2022-12-02

**Authors:** An-Qi Tan, Yun-Fei Zheng

**Affiliations:** Department of Orthodontics, Peking University School and Hospital of Stomatology, National Center of Stomatology, National Clinical Research Center for Oral Diseases, National Engineering Laboratory for Digital and Material Technology of Stomatology, No. 22, Zhongguancun South Avenue, Haidian District, Beijing 100081, China

**Keywords:** *SNHG* family, osteoblast differentiation, bone diseases, mesenchymal stem cells

## Abstract

Small nucleolar RNA host genes (*SNHGs*), members of long-chain noncoding RNAs (lncRNAs), have received increasing attention regarding their roles in multiple bone diseases. Studies have revealed that *SNHGs* display unique expression profile during osteoblast differentiation and that they could act as promising biomarkers of certain bone diseases, such as osteoporosis. Osteogenesis of mesenchymal stem cells (MSCs) is an important part of bone repair and reconstruction. Moreover, studies confirmed that the *SNHG* family participate in the regulation of osteogenic differentiation of MSCs in part by regulating important pathways of osteogenesis, such as Wnt/β-catenin signaling. Based on these observations, clarifying the *SNHG* family’s roles in osteogenesis (especially in MSCs) and their related mechanisms would provide novel ideas for possible applications of lncRNAs in the diagnosis and treatment of bone diseases. After searching, screening, browsing and intensive reading, we uncovered more than 30 papers related to the *SNHG* family and osteoblast differentiation that were published in recent years. Here, our review aims to summarize these findings in order to provide a theoretical basis for further research.

## 1. Introduction

Disruption of bone remodeling cycles is the foundation of many bone diseases, including osteoporosis [1]. In the process of bone remodeling, mesenchymal stem cells (MSCs) differentiate into osteoblasts, which then mature into osteocytes. This ability enables MSCs to participate in bone reconstruction and possess greater bone repair application potential than other cell types [2]. Small nucleolar RNAs (snoRNAs), members of noncoding RNAs, mainly exist in nucleolus and consist of about 60-300 nucleotides [3]. Our review focuses on small nucleolar RNA host genes (*SNHGs*), which are host genes of snoRNAs. In humans, more than half of snoRNAs are transcribed from host genes. Among these host genes, *SNHGs* account for about four-fifths [4]. Currently, the *SNHG* family has been reported to have more than 20 members [5].

*SNHGs*’ aberrant expression is reflected in bone disorders. It was observed that in a postmenopausal osteoporotic mouse model (OVX mice), compared with sham mice, *SNHG1* expression in serum and femoral tissue was obviously increased [6]. In serum of osteoporosis patients, *SNHG14* expression was greatly elevated [7]. In blood mononuclear cells (MNCs) from osteoporosis patients, differentiation antagonizing non-protein coding RNA (*DANCR*) was upregulated [8]. Bone marrow-derived stem cells (BMSCs) from osteoporosis patients were derived, and researchers observed a relatively high expression of *DANCR* [9]. Centofanti et al. observed aberrantly expressed long-chain noncoding RNAs (lncRNAs) in osteoblast primary cells from an osteoporosis group and control group. Among them, growth arrest specific 5 (*GAS5*) was one of the six lncRNAs with the most significant downregulation in the osteoporosis group [10].

The above findings suggest that the *SNHG* family may play a role in bone repair. In fact, studies regarding *SNHG* members and osteogenesis have increased significantly in recent years. Among *SNHG* family members, *SNHG1*, *SNHG2* (alias *GAS5*), *SNHG5*, *SNHG7*, *SNHG14* and *SNHG13* (alias *DANCR*) have been demonstrated to be involved in osteogenesis, and most of these studies focused on the relationship between *SNHG* and MSC osteoblast differentiation. Due to the complexity of their biological functions and related mechanisms, the *SNHG* family’s roles in osteogenesis have not been clearly expounded. Therefore, we try to review these functions and mechanisms here for a better summarization.

## 2. *SNHGs* Display Aberrant Expression in Osteoblast Differentiation 

Numerous studies have indicated that *SNHGs* are differentially expressed during osteoblast differentiation [6,11,12,13]. During osteogenesis induction of bone marrow mesenchymal stem cells (BMSCs), with the notable enhancement of alkaline phosphatase (ALP) activity, *SNHG1* was markedly downregulated [6]. Another study also provided evidence that *SNHG1* was downregulated in a time-dependent manner during osteogenic induction of BMSCs [13]. The above results unanimously indicate that *SNHG1* may be involved in osteogenic differentiation of BMSCs. Other *SNHG* family members also participate in osteoblast differentiation. It was observed that expression of *SNHG5* was upregulated following prolongation of BMSC osteogenic induction [11]. *SNHG7* expression was significantly reduced during osteogenesis of TNF-α-treated human dental pulp stem cells (hDPSCs). Moreover, the amplitude of *SNHG7* expression variation was related to the concentration of TNF-α in the experimental treatment [12]. 

More studies have focused on *DANCR* (alias *SNHG13*). Based on previous results showing that human amnion-derived mesenchymal stem cells (HAMSCs) may have the ability to boost osteoblast differentiation of BMSCs [14], researchers demonstrated that when BMSCs were co-cultured with HAMSCs, expression of *DANCR* was inhibited. Furthermore, *DANCR*’s downregulation may act as an intermediate process of HAMSCs’ promoting effect on BMSC osteogenesis [15]. In previous studies, which indicated that *DANCR* might influence the pathological process of osteoporosis [8], researchers assessed the expression of *DANCR* during osteoblast induction of BMSCs, and results indicated that the expression of *DANCR* gradually decreased [16].

Our team has performed a series of studies on *SNHG* members [17,18]. We found that *GAS5* displayed a gradual upregulation in hDPSCs during 14-day osteoblast induction [17]. Moreover, in our research on another *SNHG* family member, *SNHG8*, we found that *SNHG8* may be involved in periodontal ligament stem cells (PDLSCs) osteogenic differentiation. Osteogenesis of PDLSCs was induced for two days, and we found that the induction not only increased the expression of osteogenic markers but also promoted expression of *SNHG8* (Figure 1). 

*GAS5* has also received attention from other researchers. It was reported that *GAS5* expression exhibited an upward trend during osteogenic differentiation of BMSCs [19]. Furthermore, studies have observed that *GAS5* had 15 transcript variants. Although some variants’ expression was promoted during BMSC osteogenesis, the *GAS5* transcript variant 2 displayed an opposite trend [20]. 

To summarize, the studies cited above indicated that *SHNGs* exhibited different expression levels during osteoblast differentiation. During BMSC osteoblast differentiation, *SNHG1* and *DANCR* (alias *SNHG13*) tended to be downregulated, while *SNHG5* exhibited an opposite trend. During DPSC osteoblast differentiation, *SNHG7* tended to decrease. *GAS5* (alias *SNHG2*) presented a more complicated expression trend during osteoblast differentiation. More details are shown in tabular form in Table 1. These results signify that *SNHGs* might play a role in osteoblast differentiation.

## 3. The Mechanism of *SNHGs* Regulating Osteoblast Differentiation 

### 3.1. SNHG1

*SNHG1*, small nucleolar RNA host gene 1, is located on chromosome 11q12.3 in humans and includes 11 exons [21,22]. Researchers observed that *SNHG1* served as molecular sponge for *miR-181c-5p* and decrease the level of *miR-181c-5p* [6]. Previous research reported that *miR-181c-5p* might act as biomarker of osteoporosis [23]. By targeting *miR-181c-5p*, silencing of *SNHG1* enhanced osteoblast differentiation of BMSCs, while overexpression of *SNHG1* promoted osteoclast formation of BMSCs. The possible downstream molecular target of *miR-181c-5p* was SFRP1, which regulates the Wnt3a signaling pathway. This research discovered that *SNHG1* overexpression promoted SFRP1 expression but inhibited Wnt3a, while *miR-181c-5p* overexpression significantly reversed these changes. Altogether, *SNHG1* modulated the SFRP1/Wnt3a pathway via directly binding to *miR-181c-5p* and finally produced inhibiting effects on BMSC osteogenic differentiation [6]. Jiang et al. [24] also studied *SNHG1*’s role in BMSC osteogenesis. They observed that *SNHG1* inhibited p38 activation via Nedd4, which is a member of the E3 family (key enzymes of ubiquitination) [25,26,27]. MAPK signaling pathways, including p38, play important roles in osteoblast differentiation. The inactivity of p38 could induce disturbance of osteoblast differentiation [28,29]. Thus, *SNHG1* decreased p38 activity via ubiquitination, which was mediated by Nedd4, and then impeded osteoblast differentiation of BMSCs [24]. Another study in relation to BMSC osteoblast differentiation focused on Wnt/β-catenin signaling, which impeded osteoclast formation [30,31]. It was confirmed that Dickkopf 1 (DKK1) was a Wnt signaling inhibitor [32]. Via bioinformatics analysis, they found that both *SNHG1* and DKK1 interacted with *miR-101*. Later results indicated that *SNHG1* could suppress BMSC osteoblast differentiation by serving as competitive endogenous RNA (ceRNA) for *miR-101*. In this process, DKK1 was the key downstream molecular target. In short, *SNHG1* modulated Wnt/β-catenin signal pathway though the *miR-101*/DKK1 axis and finally impeded BMSC osteoblast differentiation [13].

MC3T3-E1, a pre-osteoblast cell line, is commonly used for studies regarding osteoblast induction. It was reported that during osteoblast induction of MC3T3-E1 cells, *SNHG1* played a negative role. In this process, *SNHG1* acted as a sponge for *miR-181a-5p*. Moreover, by means of online bioinformatics prediction, *PTEN* was recognized as the target of *miR-181a-5p*, and *PTEN* might participate in this modulatory axis. Overall, this study revealed that the *SNHG1/miR-181a-5p/PTEN* axis had an inhibitory effect on osteogenesis of MC3T3-E1 cells [33].

*SNHG1*’s role in osteoblast differentiation of MSCs, especially BMSCs, has been summarized thus far. In general, *SNHG1* exerts a negative effect on osteogenic differentiation. This result may help to better understand *SNHG1*-related mechanisms of bone diseases.

### 3.2. GAS5 (Alias SNHG2)

*GAS5*, growth arrest specific 5 (alias *SNHG2*), was overexpressed in growth arrest cells, hence the name growth arrest specific 5 [34]. Our team observed that silencing *GAS5* impeded osteogenic differentiation of PDLSCs, and the regulatory effect of *GAS5* was mediated through the p38/JNK pathway, which was an important pathway related to bone formation [17]. Wang et al. [19] showed that in BMSCs derived from ovariectomized (OVX) animal models, expression of *GAS5* was promoted, which indicated that *GAS5* might be involved in BMSC osteogenesis. Later experiments indicated that *GAS5* acted as a sponge for *miR-135a-5p* and decrease its level. The downstream target of *miR-135a-5p* was FOXO1, which could significantly promote osteogenic differentiation [35]. In BMSCs derived from OVX-induced osteoporotic mice, *GAS5* functioned as sponge for *miR-135a-5p* to positively modulate FOXO1 expression and finally increase osteogenesis [19]. Another study also investigated *GAS5*’s role in BMSC osteogenesis. Researchers found that *GAS5* levels were decreased in BMSCs and bone tissues of osteoporosis patients. Furthermore, *GAS5* was positively related to osteogenesis differentiation. Later studies showed that *GAS5* functioned via specifically combining with UPF1 protein. The result of this combination was the acceleration of *SMAD7*’s decay, while *SMAD7* was an inhibitor of SMAD1/5/8 [36]. In short, *GAS5* increased osteogenesis differentiation via the UPF1/*smad7* axis and then increased bone repair activity [37] (Figure 2).

LncRNAs can generate different transcript variants. Variants from the same lncRNA might display different expression profiles in diseases [38,39]. Song et al. [20] reported 15 transcript variants of *GAS5*. They focused on *GAS5* transcript variant 2 and found that it inhibited osteoblast differentiation of hBMSCs. This result was contrary to the findings of previous researches regarding *GAS5*’s roles in osteoblast differentiation of BMSCs. They further studied the mechanism underlying their findings and discovered that *GAS5* (transcript variant 2) targeted *miR-382-3p*, which could modulate *TAF1* expression. *TAF1* inhibited osteogenesis. Moreover, *TAF1* promoted *GAS5* expression, indicating that they formed a positive *GAS5/miR-382-3p/TAF1* feedback loop in hBMSCs [20].

In summary, in osteogenesis of PDLSCs and BMSCs, *GAS5* usually plays a positive role, while *GAS5* (transcript variant 2) does the opposite. More studies are necessary for a deeper understanding of *GAS5* variants’ regulatory effects in osteoblast differentiation.

### 3.3. SNHG5

*SNHG5*, small nucleolar RNA host gene 5, is 524 base pairs (bp) in length. In humans, it is located on chromosome 6q14.3 [18,40]. It was observed that knockdown of *SNHG5* impeded hBMSCs osteoblast differentiation. The mechanism underlying this phenomenon was related to *miR-582-5p*. *SNHG5* served as sponge for *miR-582-5p*. As a result, *SNHG5* modulated the expression of RUNX3. RUNX3 belongs to the RUNX family and has been confirmed to function as an inducer of osteogenic differentiation. Mechanically, *SNHG5* served as sponge for *miR-582-5p* and then modulated RUNX3 expression to promote hBMSC osteogenic differentiation. Moreover, RUNX3 also promoted activity of *SNHG5* at the level of transcription. In this way, a positive feedback loop of *SNHG5/miR-582-5p*/RUNX3 was formed [11].

Our team also studied the positive role of *SNHG5* in the osteogenesis and identified a regulatory axis that modulates hBMSC osteogenic differentiation: YY1/*SNHG5/miR-212-3P/GDF5*/Smad. Through this axis, promoting *SNHG5* expression stimulated hBMSC osteogenic differentiation, which might provide a potential target for bone repair [18]. It is worth mentioning that in another study on periodontitis, our team found that local si-*SNHG5* injection into mouse periodontitis model resulted in severe bone loss [41]. The positive role of *SNHG5* in bone formation was further demonstrated by this finding.

### 3.4. SNHG7

*SNHG7*, small nucleolar RNA host gene 7, is 2176 bp in length. In humans, *SNHG7* is located in the chromosome 9q34.3 [42,43]. It was reported that in the pre-osteoblast cell line, MC3T3-E1, *SNHG7* functioned as sponge for *miR-9*, while *miR-9* targeted the 3’UTR of TGFBR2. Further study revealed that knockdown of *SNHG7* inhibited the TGF-β signaling pathway. TGF-β has been confirmed to act as significant regulator of osteoblast and osteoclast differentiation [44]. These results showed that knockdown of *SNHG7* resulted in lower osteogenic activation via the *miR-9*/TGF-β axis [45]. Additionally, in TNF-α-treated hDPSCs, research reported that *SNHG7* exerted a positive effect on osteoblast differentiation [12]. In this research, hDPSCs were treated with 50 ng/mL of TNF-α to simulate an inflammatory microenvironment, and these cells then underwent osteogenic induction. Researchers observed that in hDPSCs, promoting expression of *SNHG7* reversed the inhibiting influence of TNF-α on osteoblast/dentinogenic differentiation. Additionally, *miR-6512-3p* participated in this process via interacting with *SNHG7*. In short, *SNHG7* promoted osteoblast differentiation of hDPSCs treated with TNF-α via targeting *miR-6512-3p* [12]. Another study also showed that silencing of *SNHG7* impeded the odontogenic and osteogenic activity of hDPSCs [46]. Researchers identified aberrantly expressed mRNAs, lncRNAs and microRNAs during osteo/odontogenic induction of DPSCs. Among them, *SNHG7* was one of the most upregulated. To verify *SNHG7*’s role in osteo/odontogenic induction of DPSCs, *SNHG7* silencing was conducted, and as a result, the expression of mineralization and osteogenic marker genes in DPSCs was inhibited [46]. These results are consistent with the positive role of *SNHG7* in osteogenesis shown in previous studies.

### 3.5. DANCR (Alias SNHG13)

*DANCR*, differentiation antagonizing non-protein coding RNA, is found on chromosome 4q12 [47]. As a member of lncRNA family, *DANCR* participates in numerous diseases and pathological process, such as tumors and bone diseases [48,49]. There have been an increasing number of studies focusing on the relationship between *DANCR* and osteoblast differentiation regulation. For example, Tang et al. [50] showed that *DANCR* participated in osteolysis after total hip arthroplasty, and in a subsequent experiment, they found that DANCR could decrease MSCs’ osteogenesis activity via modulating FOXO1, which is a transcription factor (Figure 3).

Additional studies have focused on the relationship between *DANCR* and BMSC differentiation. Previous studies have confirmed that human amnion-derived mesenchymal stem cells (HAMSCs) had the ability to stimulate osteoblast differentiation of hBMSCs [51]. To explore the mechanism, researchers focused on the differentially expressed lncRNAs in hBMSCs co-cultured with HAMSCs. Among these differentially expressed genes, *DANCR* attracted attention of researchers. They observed that promoting *DANCR* expression impeded osteoblast differentiation of hBMSCs in the co-culture system [15]. Another study also showed that *DANCR* exerted inhibitory effects on hBMSC osteoblast differentiation. It was reported that *DANCR* modulated hBMSC osteoblast differentiation by absorbing *miR-1301-3p*. Moreover, they confirmed that PROX1, a transcription factor, was a downstream target of *miR-1301-3p*. In summary, this study showed that *DANCR* inhibited hBMSC osteogenesis via the *DANCR/miR-1301-3p*/PROX1 axis [16]. Additionally, it was noticed that in BMSCs derived from osteoporosis patients, *DANCR* expression was decreased during osteoblast induction. This research first verified the interaction between *DANCR*, *miR-320a* and *CTNNB1* during osteogenesis. It was demonstrated that *DANCR* and *miR-320a* did not interact with each other in terms of expression. *CTNNB1*, the gene that encodes β-catenin, was directly modulated by *miR-320a*. In short, this study verified a new network composed of *DANCR*, *miR-320a* and *CTNNB1* through the Wnt/β-catenin pathway during hBMSC osteoblast differentiation [9]. 

In MC3T3-E1, a pre-osteoblast cell line, *DANCR* also attracted researchers’ attention. After silencing *DANCR* in MC3T3-E1 cells, researchers observed an increasing trend of osteoblast differentiation. Further study revealed that the Wnt/β-catenin pathway also participated in this effect, which is consistent with abovementioned results [52]. 

Additionally, effects of nutrients on osteogenesis also related to *DANCR*. Yang et al. [47] reported that sesamin displayed a dual function in both osteogenesis promotion and osteoclastogenesis inhibition. Sesamin increased osteogenesis by activating the Wnt/β-catenin pathway and decreased osteoclastogenesis via deactivating the NF-κB signaling pathway. Previous studies confirmed that deactivating the NF-κB signaling pathway inhibited osteoclast formation [53]. Further studies revealed that sesamin’s promoting effects on osteoblast were attributable to downregulation of *DANCR* expression. In other words, *DANCR* exhibited negative effects on osteoblasts, which was consistent with the findings of previous studies [9,16,54].

### 3.6. SNHG14

*SNHG14*, small nucleolar RNA host gene 14, has been scarcely reported in relation to its effects on the differentiation of MSCs. In humans, *SNHG14* is located on chromosome 15q11.2 [55]. Researchers noticed that *SNHG14* may show disordered expression during human mesenchymal stem cells (hMSCs) differentiation. Further studies showed that silencing of *SNHG14* inhibited osteogenesis of hMSCs via the *miR-2861*/AKT2 axis. More specifically, the promoting effect of *SNHG14* on hMSC osteogenesis could be reversed by *miR-2861*, and the downstream target of *miR-2861* was AKT2 [7]. Consequently, promoting *SNHG14* expression would ultimately stimulate hMSC osteoblast differentiation.

## 4. Prospect

In recent years, *SNHGs* are gaining increased attention in the epigenetics of bone remodeling. Researchers in related fields focus on relationships between the *SNHG* family and osteogenesis, and the purpose of this review was to summarize findings of this research field (Table 2).

Our team has found that the *SNHG* family not only plays a role in osteogenesis but also participates in the progression of inflammation. For example, we found that *SNHG5* mediated periodontal inflammation via the NF-κB signaling pathway [41]. Another *SNHG* family member, *SNHG8*, also affected the NF-κB signaling pathway and inflammatory processes [53]. We observed that in a hypoxic inflammatory microenvironment, decreased *SNHG8* expression activated the downstream NF-κB signaling pathway. This indicated that *SNHG8* participated in periodontal tissue reconstruction, which could be related to the bone remodeling discussed in this review. It can be considered that the *SNHG* family not only affects bone reconstruction through osteogenesis but also through its effects on inflammation. This conclusion provides novel insights for exploring more functions of the *SNHG* family.

Understanding of the *SNHG* family’s regulatory network in osteoblast differentiation remains to be improved. Most of studies mentioned above were based on ceRNA (competing endogenous RNA) mechanisms. *SNHGs* could act as ceRNAs and competitively bind to specific microRNAs. Downstream, microRNA mediates the silencing of specific genes [56]. However, in several studies mentioned in this review, *SNHGs* do not function through ceRNA mechanisms but directly bind to proteins (Figure 2 and Figure 3). This suggests that the ceRNA mechanism involved in *SNHG* function may have a parallel protein mechanism. Moreover, there may be an intersection between these two kinds of mechanisms. In general, *SNHGs*’ roles in osteogenesis tend to be diverse, and more work needs to be conducted to establish definitive mechanisms through which *SNHGs* modulate osteoblast differentiation. 

## 5. Conclusions 

In this review, we first summarized *SNHGs* that reportedly participate in osteogenic differentiation, especially in MSCs. To summarize, *SNHG1* and *SNHG 13* (alias *DANCR*) have negative effects on osteogenesis, while *SNHG5*, *SNHG7* and *SNHG14* can promote osteogenesis. Except the inhibitory effect of *GAS5* transcript variant 2 on osteogenesis, most studies of *GAS5* (alias *SNHG2*) show that it can promote osteogenesis.

In addition, we discussed their molecular regulatory mechanisms, which can be divided into two types: lncRNA-miRNA-mRNA crosstalk and direct interaction with protein. In the future, researchers may reveal more molecules which are involved in the *SNHG* family’s regulatory network and further verify the *SNHG* family’s effects on osteoblast differentiation.

Regarding the treatment of bone diseases, for the needs of bone regeneration, MSCs have always been the focus of researchers, and MSC implantation has shown its clinical value in the therapy of osteoporosis [57]. Moreover, adding specific growth factors could improve the treatment effect [58]. Most of studies on *SNHGs* and osteogenesis were carried out in MSCs, and the effects of *SNHGs* on MSC osteogenesis could have potential to be applied in clinic. For example, based on the existing MSC implantation therapy, it could be considered to use *SNHGs*’ effects at the same time to increase the therapeutic effect. We expect that the above findings regarding the *SNHG* family and osteogenesis will eventually guide researchers to find new treatments for bone diseases.

## Figures and Tables

**Figure 1 genes-13-02268-f001:**
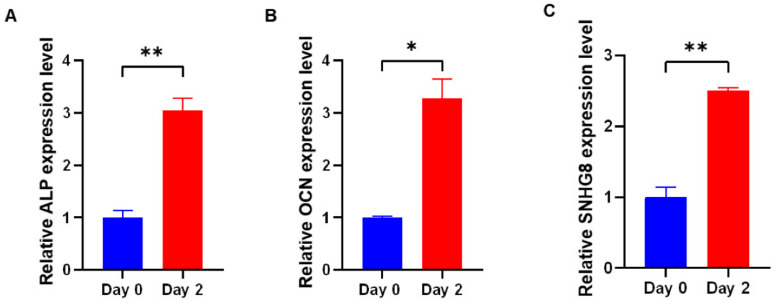
The expression of *SNHG8* increased in osteogenic medium. For the induction of osteoblast differentiation, PDLSCs were cultured in osteogenic medium (OM), which was composed of growth medium supplemented with b-glycerophosphate (10 mM), dexamethasone (100 nM) and vitamin C (200 mM). The relative mRNA expression of (**A**) alkaline phosphatase (ALP), (**B**) osteocalcin (OCN) and (**C**) small nucleolar RNA host gene 8 (*SNHG8*) was measured by qRT-PCR on the second day of osteogenic induction. Three repeated experiments were conducted. Data are presented as the mean ± SD, * *p* < 0.05, ** *p* < 0.01.

**Figure 2 genes-13-02268-f002:**
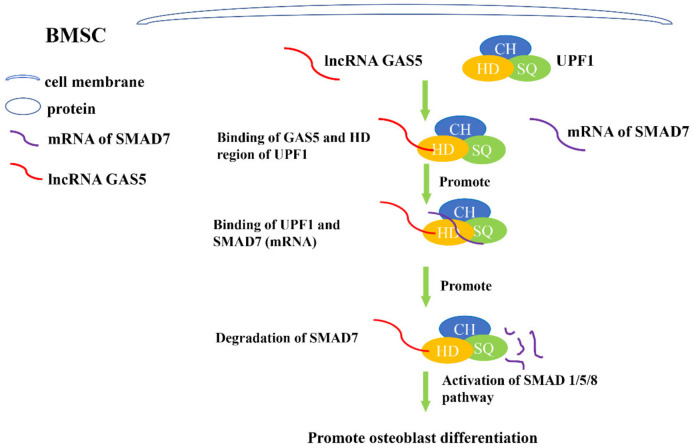
Schematic illustration of lncRNA *GAS5* regulating osteogenesis of BMSCs by directly binding to protein UPF1. *GAS5* binds to the HD region of UPF1 specifically, which results in binding of UPF1 and *SMAD7* (mRNA). After binding to UPF1, *SMAD7* (mRNA) degrades. As an inhibitor of the SMAD 1/5/8 pathway, degradation of *SMAD7* promotes activation of SMAD 1/5/8 pathway and ultimately promotes osteoblast differentiation of BMSCs.

**Figure 3 genes-13-02268-f003:**
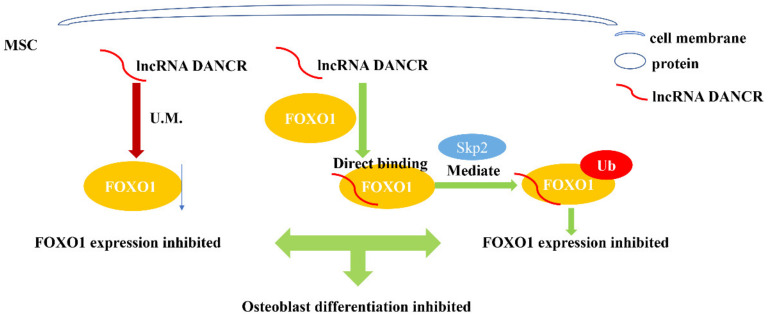
Schematic illustration of lncRNA *DANCR* regulating osteoblast differentiation of MSC by directly binding to protein FOXO1. Through unknown mechanisms, *DANCR* inhibits FOXO1 expression. Additionally, *DANCR* directly binds to FOXO1 and promotes Skp2-mediated ubiquitination of FOXO1, which further reduces FOXO1 expression. FOXO1, a transcription factor, regulates proliferation, differentiation and apoptosis of osteoblasts. By decreasing FOXO1 expression, *DANCR* ultimately inhibits osteoblast differentiation of MSC. U.M., unknown mechanism.

**Table 1 genes-13-02268-t001:** *SNHG* expression trends during osteoblast differentiation.

LncRNA	Cell	Expression Trend during Osteoblast Differentiation	References
** *SNHG1* **	BMSC	Downregulation	[6,13]
** *SNHG2 (GAS5)* **	hDPSC/BMSC	Upregulation	[17,19]
***GAS5*** transcript variant 2	hBMSC	Downregulation	[20]
** *SNHG5* **	hBMSC	Upregulation	[11,18]
** *SNHG7* **	hDPSC	Downregulation	[12]
***SNHG13*** (***DANCR***)	hBMSC (co-cultured with HAMSC)	Downregulation	[15]
***SNHG13*** (***DANCR***)	BMSC	Downregulation	[16]

**Table 2 genes-13-02268-t002:** Key *SNHGs* regulate osteoblast differentiation.

LncRNA	Cell	Regulatory Axis	LncRNA’s Effect on Osteogenesis	References
** *SNHG1* **	BMSC	*SNHG1/miR-181c-5p*/SFRP1/Wnt3a	Inhibitory	[6]
** *SNHG1* **	BMSC	*SNHG1*/Nedd4/p38	Inhibitory	[24]
** *SNHG1* **	BMSC	*SNHG1/miR101*/DKK1/Wnt/ β-catenin	Inhibitory	[13]
** *SNHG1* **	MC3T3-E1	*SNHG1/miR-181a-* *5p/PTEN*	Inhibitory	[33]
** *SNHG2 (GAS5)* **	hPDLSC	*SNHG2*/p38/JNK	Stimulative	[17]
** *SNHG2 (GAS5)* **	BMSC	*SNHG2/miR-135a-5p*/FOXO1	Stimulative	[19]
** *SNHG2 (GAS5)* **	BMSC	*SNHG2*/UPF1/*Smad7*	Stimulative	[37]
***GAS5*** transcript variant 2	BMSC	*SNHG2/miR-382-3p/TAF1*	Inhibitory	[20]
** *SNHG5* **	BMSC	*SNHG5/miR-582-5p*/RUNX3	Stimulative	[11]
** *SNHG5* **	BMSC	YY1/*SNHG5/miR-212-3p/GDF5*/Smad	Stimulative	[18]
** *SNHG7* **	MC3T3-E1	*SNHG7/miR-9*/TGF-β	Stimulative	[45]
** *SNHG7* **	hDPSC	*SNHG7/miR-6512-3p*/TNF-α	*Stimulative*	[12]
** *SNHG13 (DANCR)* **	MSC	*SNHG13*/FOXO1	Inhibitory	[50]
** *SNHG13 (DANCR)* **	BMSC	*SNHG13/miR-1301-3p*/PROX1	Inhibitory	[16]
** *SNHG13 (DANCR)* **	BMSC	*SNHG13/miR-320a/CTNNB1*/Wnt/β-catenin	Inhibitory	[9]
** *SNHG13 (DANCR)* **	MC3T3-E1	*SNHG13*/Wnt/β-catenin	Inhibitory	[52]
** *SNHG14* **	hMSC	*SNHG14/miR-2861*/AKT2	Stimulative	[7]

## Data Availability

Not applicable.

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
