# Peer review of "The Roles of SNHG Family in Osteoblast Differentiation"

_genes, 2022, doi:10.3390/genes13122268_

Round 1
Reviewer 1 Report
I have extensively reviewed the article entitled, “The roles of SNHG family in osteoblast differentiation of mesenchymal stem cells”. The authors have thoroughly done review of literature in order to summarize the main findings in the field. I would suggest a major revision which will need significant amount of changes in overall manuscript.
Comments:
1. Title: The title of this review article reflects the role of SNHG family in the osteogenic differentiation of MSCs. However, there are some sections that are not best fit under the present title for example section 3 highlights their role in osteoporosis and other bone diseases, section 4.7. bone remodeling
The authors are advised to either change the title of the article or try to keep the MSCs relevant sections only.
2. Abstract: The abstract is very vague written, the authors should cover three components in the abstract
a) Background; highlighting the need for this review
b) Importance of the topic in the field
c) Authors approach
3. Introduction: Authors should start the introduction by highlighting the overall field, for example why is osteogenesis of MSCs required in bone regeneration etc. Followed by why are MSCs preferable over other cells. Then introducing the SNHG family and their role in MSC osteogenic differentiation.
Spelling mistake
Line 10 mall nucleolar, it should be small nucleolar
4. Section 2, LNC-SNHGs display aberrant expression in osteoblast differentiation of MSCs
The overall section is very vaguely written, I am highlighting few points
a) The authors have started the section by stating that, ‘Numerous researchers have indicated…… There should be multiple references on this statement
b) Line 60…… process of HAMSCs controlling.
The authors should be specific about what controlling
c) This section is about MSCs, in line 61 suddenly a statement about osteoporosis is put in the section, which does not make any sense. Authors should correlate this statement with MSCs.
d) Line 65 states that “Our team has performed a series of researches…. This statement should also be supported by multiple references.
e) Figure 1, is it taken from any previously published work that should be cited or if it is reported for the first time then some experimental background should be added, for instance how the osteogenesis was induced in these MSCs.
5. Section 3 LNC-SNHGs display aberrant expression in osteoporosis and other bone diseases
a) This section does not highlight the role of SNHGs and MSCs in correlation to bone diseases.
b) This section will be best fit if authors try to describe the role or SNHGs in MSCs to combat the bone diseases or their role in bone diseases.
c) Other than osteoporosis, authors have summarized the studies related to fractures and hip replacement, the later two are not considered as bone diseases, author should change the title of this section related to bone disorders
d) Also summarize the role in bone disease for example, osteomyelitis, osteosarcoma, metastatic bone cancers etc.
e) The sentence, “Among them, GAS5 was one of the six lncRNAs with the most significant downregulation” is incomplete. Downregulation in the control or osteoporotic group?
The overall section 3 should be mentioned in in abstract as well
6. Section 4.1
a) Paragraph 2 is all about MC3T3-E1 (mouse pre-osteoblasts), these are not MSCs and hence should not be included in this article.
b) Song et al. [21] reported 15 transcript variants of GAS5. They focused on GAS5 177 transcript variants 2
Authors should add why was variant 2 focused, does it have any specific role or how does the cited study decide to choose variant 2 over all other varients
7. Section 4.6.
a) line 247., which type of tumors, what is the actual role of DANCR in those tumors
b) The last paragraph from line 276 and 283, There is no correlation of the whole paragraph with the section because the authors stated that the effects of nutrients on osteogenesis also related to DANCR……..revealed that sesamin’s promoting effects on osteoblast were realized by downregulating DANCR expression. In other words, DANCR exhibited negative effects on osteoblast, which was consistent with previous studies’ findings.
There is no reference for this statement.
8. Why are authors discussing SNHG14 before SNHG13, is there any order followed or based on the importance?
9. Section 4.7 SNHG family and bone remodeling in other cell types
a) This section is not a good fit under the present title of the article
b) If the article title is about MSCs then the studies involving osteoclasts or preosteoblasts should not be included unless the cited studies have reported any effect on MSCs.
10. Section 5. Prospect
a) Only summarizing the studies does not suffice the need to write the review article, it should rather build ideas and authors should have their take on the reviewed field, which should be added in this section.
b) ceRNA has been added abruptly here, no connectivity with the major portion of the review article
Section 6. Conclusion
Conclusion is vague, it should reflect the authors statements based upon the cited literature rather than repeating the same statements as in the major section of the manuscript
The authors should add concluding remarks rather than a general statement as the last paragraph
11. Every acronym needs to be spelled out when used at the first place of mention. For example PDSCs, PDLCs, BMSCs, ALP etc.
12. The abbreviations using the Greek alphabets should not use English alphabets for example, TNF-a should be replaced with TNF-α
13. Line 242. …. the downstream molecular of miR-2861 is AKT2.
It should be either
… downstream molecule of… or … downstream molecular target of…
Author Response
Dear Reviewers,
We are very grateful because you have spent a substantial amount of time looking over our manuscript entitled “The roles of SNHG family in osteoblast differentiation” and raised insightful comments.
We have addressed the questions point by point in the word.

Reviewer 2 Report
Dear Authors
the paper is very interesting and minor revisions are needed.
Indeed, some topics about potential clinical relevance of your results need to be inserted:
1) Such results can help us to know the quality of bone in diabetic patients to get better results for osseointegration? Please cite DOI10.3390/ijerph191811735
2) Such results can help the clinician to get better results during the follow-up of peri-implant bone ? Please cite DOI10.23805/JO.2018.10.04.04
3) Such results can help in the management of tooth orthodontic movements? Please cite DOI10.1177/1721727X1201000208
Author Response

(The authors gave the same response as above.)

Reviewer 3 Report
This is an interesting review by Tan et al where they discuss the role of SNHG family in osteoblast differentiation. This study is exciting and provides overview of involvement of different SNHG in osteoblast differentiations, which can be improved with minor changes to the manuscript.
Minor Comments:
1. There are some typos error and some abbreviation is missing that’s need to be corrected.
2. Rephrase the line number 25-26 to better understanding for the reader.
3. In the figure 1 author mentioned only about the osteogenic medium, describe the term with respect to composition for better readability and understanding.
4. In conclusion section author discuss about how SNHGs regulates the osteogenesis, and its further applications, such as biomarkers and new therapeutic targets development of bone diseases, including osteoporosis. In accordance to this better to describe about the current existing therapies.
Author Response

(The authors gave the same response as above.)

Round 2
Reviewer 1 Report
The revised manuscript with the track changes is very confusing, authors are requested to make the track changes only to the sections/sentence where it is required for example the track changes in the abstract are well accepted. However, from introduction onwards, authors have made track changes even to those sentences, where no changes are made. For example first paragraph of introduction (as highlighted below) is kept in track changes, however there is no change as compared to the previous.
Disorder of bone remodeling cycle is the foundation of many bone diseases, including osteoporosis [ 1]. In the progress of bone remodeling, mesenchymal stem cells (MSCs) could differentiate into osteoblasts, which then mature into osteocytes. This ability enables MSCs to participate in bone reconstruction and possess greater bone repair application potential than other cell types [ 2].
Similarly, DANCR (alias SNHG13) section
DANCR, differentiation antagonizing non-protein coding RNA, presents on Chromosome 4q12 [47]. As a member of lncRNA family, DANCR participates in numbers of diseases’ pathological process, such as tumors, bone diseases [48,49]. There have been more and more researches pay attention to relation between DANCR and osteoblast differentiation regulation. For example, Tang et al. [50] observed DANCR participated in osteolysis after total hip arthroplasty, and in subsequent experiment, they found that DANCR could decrease MSCs’ osteogenesis activity via modulating FOXO1, which was a transcription factor (Figure 3).
Similarly there are more track-changes like these, which are actually similar to the previous manuscript, hence should be kept in plain text.
Track changes should only reflect the changes which are made in the revised manuscript.
In present format, the revised manuscript is very confusing to read
Please revise by only keeping those sentences in track changes, where changes are made.
Also authors have mentioned in the cover letter that changes in the title are made but in the revised manuscript the title is the same as previous one.
Author Response
Thanks for your comments. We submit the final version of manuscript with no track of change last time. Because the last revised manuscript is very different from the manuscript submitted for the first time, it involves a lot of content movement and deletion, so we submitted the final version of manuscript with no track of change last time. we suppose that you may have read a different version of manuscript, or there are some misunderstandings. And thank you for the uncorrected negligence of the title. We revised the manuscript again, and uploaded the final manuscript with no revision marks (no track of changes) this time.